# TequilaGAN: How to easily identify GAN samples

## Abstract

In this paper we show strategies to easily identify fake samples generated with the Generative Adversarial Network framework. One strategy is based on the statistical analysis and comparison of raw pixel values and features extracted from them. The other strategy learns formal specifications from the real data and shows that fake samples violate the specifications of the real data. We show that fake samples produced with GANs have a universal signature that can be used to identify fake samples. We provide results on MNIST, CIFAR10, music and speech data.

## 1 Introduction

*Fake samples* generated with the Generative Adversarial Networks (Goodfellow et al., 2014a) framework have fooled humans and machines to believe that they are indistinguishable from real samples. Although this might be true for the naked eye and the discriminator fooled by the generator, it is unlikely that fake samples are numerically indistinguishable from real samples. Inspired by formal methods, this paper focuses on the evaluation of fake samples with respect to statistical summaries and formal specifications computed on the real data.

Since the Generative Adversarial Networks paper (Goodfellow et al., 2014a) , most GAN related publications use a grid of image samples to accompany theoretical and empirical results. Unlike Variational Autoencoders (VAEs) and other models (Goodfellow et al., 2014a), most of the evaluation of the output of GAN trained Generators is qualitative: authors normally list higher sample quality as one of the advantages of their method over other methods. Although numerical measures like the inception score are used to evaluate GAN samples (Salimans et al., 2016), interestingly, little is mentioned about the numerical properties of fake samples and how these properties compare to real samples.

In the context of Verified Artificial Intelligence (Seshia & Sadigh, 2016), it is hard to systematically verify that the output of a model satisfies the specifications of the data it was trained on, specially when verification depends on the existence of perceptually meaningful features. For example, consider a model that generates images of humans: although it is possible to compare color histograms of real and fake samples, we do not yet have robust algorithms to verify if an image follows specifications derived from anatomy.

This paper is related to the systematic verification of fake samples and focuses on comparing numerical properties of fake and real samples. In addition to comparing statistical summaries, we investigate how the Generator approximates modes in the real distribution and verify if the generated samples violate specifications derived from it. We offer the following main contributions:

- We show that fake samples have properties that are barely noticed with visual inspection
- We show that these properties can be used to identify the source of the data (real or fake)
- We show that fake samples violate formal specifications learned from real data

## 2 Related work

Despite its youth, several publications ((Arjovsky & Bottou, 2017), (Salimans et al., 2016), (Zhao et al., 2016), (Radford et al., 2015)) have investigated the use of the GAN framework for sample

generation and unsupervised feature learning. Following the procedure described in (Breuleux et al., 2011) and used in (Goodfellow et al., 2014a), earlier GAN papers evaluated the quality of the fake samples by fitting a Gaussian Parzen window[1] to the fake samples and reporting the log-likelihood of the test set under this distribution. As mentioned in (Goodfellow et al., 2014a), this method has some drawbacks, including its high variance and bad performance in high dimensional spaces. The inception score is another widely adopted evaluation metric that fails to provide systematic guidance on the evaluation of GAN models(Barratt & Sharma, 2018).

Unlike other optimization problems, where analysis of the empirical risk is a strong indicator of progress, in GANs the decrease in loss is not always correlated with increase in image quality (Arjovsky et al., 2017), and thus authors still rely on visual inspection of generated images. Based on visual inspection, authors confirm that they have not observed mode collapse or that their framework is robust to mode collapse if some criteria is met ((Arjovsky et al., 2017), (Gulrajani et al., 2017), (Mao et al., 2016), (Radford et al., 2015)). In practice, github issues where practitioners report mode collapse or not enough variety abound.

In their publications, (Mao et al., 2016), (Arjovsky et al., 2017) and (Gulrajani et al., 2017) propose alternative objective functions and algorithms that circumvent problems that are common when using the original GAN objective described in (Goodfellow et al., 2014a). The problems addressed include instability of learning, mode collapse and meaningful loss curves (Salimans et al., 2016).

These alternatives do not eliminate the need, or excitement[2], to visually inspect GAN samples during training, nor do they provide quantitative information about the generated samples.

## 3 METHODOLOGY

The experiments in this paper focus on three points: the first shows that fake samples have properties that are hardly noticed with visual inspection and tightly related to the requirements of differentiability; the second shows that there are numerical differences between statistical moments computed on features extracted from real and fake samples that can be used to identify the data; the third shows that fake samples violate formal specifications learned from the real data. In the following subsections we enumerate the datasets, features and GAN frameworks herein used.

### 3.1 DATASETS

In our experiments, we use MNIST, CIFAR10, a MIDI dataset of 389 Bach Chorales downloaded from the web and a subsample of the NIST 2004 speech dataset used in (Cai et al., 2018).

### 3.2 FEATURES

The **spectral centroid** (Peeters, 2004) is a feature commonly used in the audio domain, where it represents the barycenter of the spectrum. This feature can be applied to other domains and we invite the reader to visualize Figure 12 for examples on MNIST and Mel-Spectrograms (Peeters, 2004). For each column in an image, we transform the pixel values into row probabilities by normalizing them by the column sum, after which we take the expected row value, thus obtaining the spectral centroid.

The **spectral slope** adapted from (Peeters, 2004) is computed by applying linear regression using an overlapping sliding window of size 7. For each window, we regress the spectral centroids on the column number *mod* the window size. Figure 13 shows these features computed on MNIST and Mel-Spectrograms.

### 3.3 GAN FRAMEWORKS

We investigate samples produced with the DCGAN architecture using the Least-Squares GAN (LSGAN) (Mao et al., 2016) and the improved Wasserstein GAN (IWGAN/WGAN-GP) (Gulrajani et al., 2017). We also compare adversarial MNIST samples produced with the fast gradient sign

---

[1]Kernel Density Estimation

[2]Despite of authors promising on Twitter to never train GANs again.

method (FGSM) (Goodfellow et al., 2014b). We evaluate the normally used non-linearities, sigmoid and tanh, on the output of the generator and other transformations such as the scaled tanh and identity.

# 4 EXPERIMENTS

## 4.1 MNIST

This experiment focuses on showing numerical properties of fake MNIST samples and features therein, unknown to the naked eye, that can be used to identify them as produced by a GAN.

We start by comparing the distribution of features computed over the MNIST training set to other datasets, including the MNIST test set, samples generated with GANs and adversarial samples computed using the FGSM. The training data is scaled to $[0, 1]$ and the random baseline is sampled from a Bernoulli distribution with probability equal to the mean value of pixel intensities in the MNIST training data, 0.13. Each GAN model is trained until the loss plateaus and the generated samples look similar to the real samples. The datasets compared have 10 thousand samples each.

Visual inspection of the generated samples in Figure 14 show that IWGAN seems to produce better samples than LSGAN. Quantitatively, we use the MNIST training set as a reference and compare the distribution of pixel intensities. Table 1 reveals that although samples generated with LSGAN and IWGAN look similar to the training set, they are considerably different from the training set given the Kolgomorov-Smirnov (KS) Two Sample Test and the Jensen-Shannon Divergence (JSD), specially with respect to the same statistics on the MNIST test data.

|  | KS Two Sample Test | | JSD |
|---|---|---|---|
|  | Statistic | P-Value | |
| mnist_train | 0.0 | 1.0 | 0.0 |
| mnist_test | 0.003177 | 0.0 | 0.000029 |
| mnist_lsgan | 0.808119 | 0.0 | 0.013517 |
| mnist_iwgan | 0.701573 | 0.0 | 0.014662 |
| mnist_adversarial | 0.419338 | 0.0 | 0.581769 |
| mnist_bernoulli | 0.130855 | 0.0 | 0.0785009 |

Table 1: Statistical comparison over the distribution of pixel values for different samples using MNIST training set as reference.

These numerical phenomena can be understood by investigating the empirical CDFs in Figure 1. The distribution of pixel values of the samples generated with the GAN framework is mainly bi-modal and asymptotically approaches the modes of the distribution in the real data, values 0 and 1. Expectedly, the FGSM method, noted as *mnist_adversarial*, causes a shift on the modes of the distribution that can be easily identified.

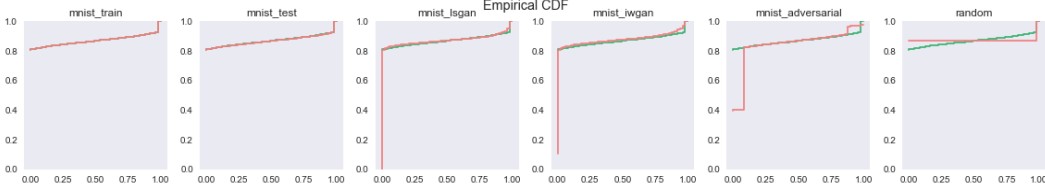

Figure 1: Pixel empirical CDF of training data as reference (green) and other datasets(red)

In addition, plots of the distribution of statistical moments of the spectral centroid in 2 suggests that the fake images are more noisy than the real images. Consider for example images produced by randomly sampling a Bernoulli distribution with parameter estimated from the training data. These images have pixel values of 0 or 1 that are equally distributed [3] over the image. Well, an image that has pixels values distributed in such a manner will have a distribution of mean spectral centroid with a mode at the center row of the image. This and the similarity between the distribution of mean

---

[3]This spatial distribution is independent of the parameter of the Bernoulli distribution.

spectral centroids from fake data and Bernoulli data suggest that the fake images have noise that are also equally spatially distributed.

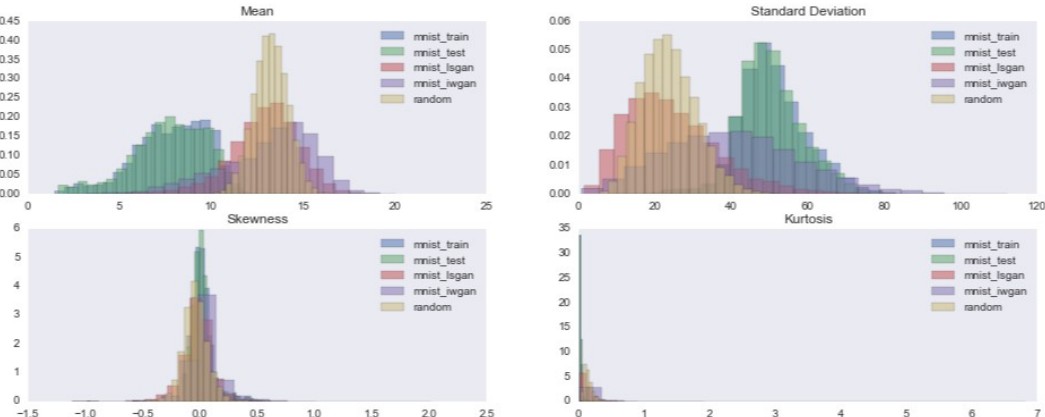

Figure 2: Distribution of moments of spectral centroids computed on each image.

Last, Figure 3 shows that the GAN generated samples smoothly approximate the modes of the distribution. This smooth approximation is considerably different from the training and test sets. Although not perceptually meaningful, these properties can be used to identify the source of the data.

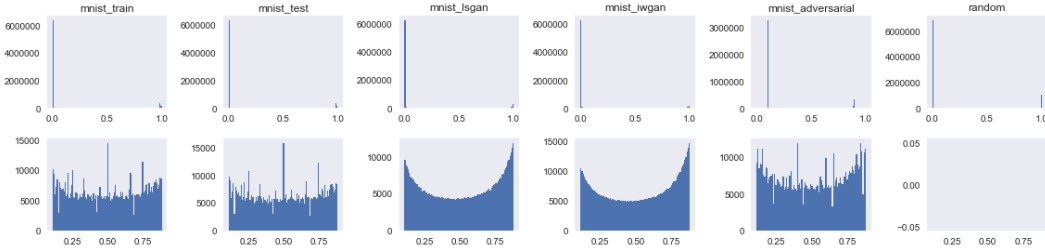

Figure 3: Histogram of pixel values for each dataset. First row shows values within [0, 1] and 100 bins. Second row shows values within [0.11, and 0.88] and 100 bins.

Our first hypothesis for the smooth approximation of the modes of the distribution was that it would be present in any data produced with a generator that is trained using stochastic gradient descent **and** a saturating activation function, such as sigmoid or tanh, at the output of the generator. To evaluate this hypothesis, we conducted a set of experiments using different GAN architectures (WGAN-GP, LSGAN, DCGAN) with different activation functions, including linear and the scaled tanh, at the output of the Generator, keeping the Discriminator fixed.

To our surprise, we noticed that the models trained with linear or scaled tanh activations were partially able to produce images that were similar to the MNIST training data and the distribution of pixel intensities, although uni-modal around zero, still possessed a smooth looking curve. This is illustrated in Figure 4.

We then postulated that the smooth behavior was due to smoothness in the pixels intensities of the training data itself. To evaluate this, we binarized the real data by first scaling it between $[0, 1]$ and then thresholding it at $0.5$. With this alteration the distribution of the pixel intensities of the real data becomes completely bi-modal with modes at $0$ and $1$. Figure 5 shows that the smooth behavior remained.

With this empirical evidence at hand, we provide an informal analysis of the smoothness of the distribution of pixels of the generated data from the perspective of optimization, differentiation and function approximation with neural networks. We know that backpropagation and stochastic gradient descent are used to update the weights of a neural network model based on the gradient of the loss with respect to the weights. We also know that differentiation requires the function that is being

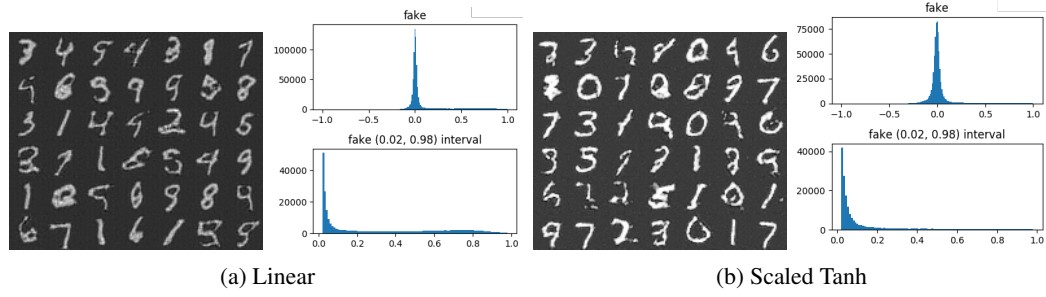

(a) Linear                            (b) Scaled Tanh

Figure 4: Fake MNIST samples and pixel distribution from generators trained with DCGAN, Batch Norm and linear or scaled tanh activation functions.

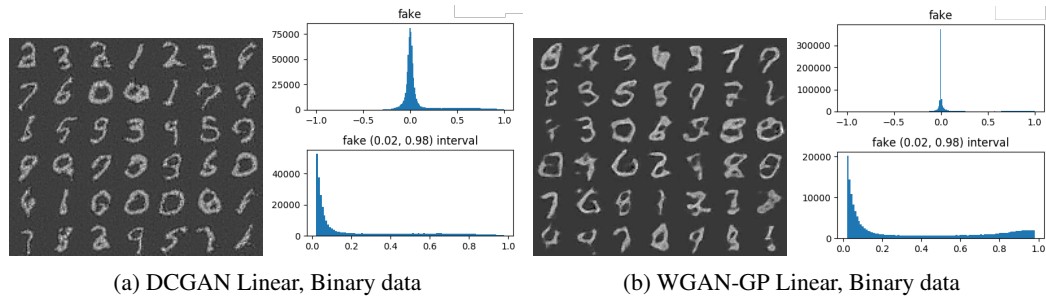

(a) DCGAN Linear, Binary data            (b) WGAN-GP Linear, Binary data

Figure 5: Fake MNIST samples and pixel distribution from generators trained on binarized real data with DCGAN and WGAN-GP, Batch Norm and linear activation functions.

differentiated to have some high degree of smoothness and be differentiable almost everywhere. Hence, we conjecture that the inductive bias of this learning setup is that of smoothness given the requirements of differentiation.

We also speculate that this inductive bias is responsible for the smoothness of the distribution of pixel values at any iteration during training and that the U shape of the distribution of pixel values, similar to blurring, is the byproduct of an smooth approximation of the function that is being learned[4].

## 4.2 CIFAR10

Expecting that the results we obtained during our MNIST experiments would generalize to other images, we briefly investigate the properties of CIFAR10 fake samples generated with the IWGAN framework and using the DCGAN architecture. The models were trained using the CIFAR10's training set with 50k samples and following the experimental setup provided by the IWGAN authors.

We report results on CIFAR10 train, test and IWGAN generated samples, all with ten thousand items each. An informal analysis of Figure 6a shows that the distribution of pixels per channel is different between the real and fake data, specially for pixels values close to $-1$. Numerically, we can see in Table 6b that the JSD between the samples from the training data and IWGAN samples is considerably large with respect to the same statistics on the test data.

Figure 7 shows a behavior seen in our MNIST experiments: the GAN generated samples smoothly approximate the mode of the pixel value distribution at 1 and this smooth approximation is considerably different from the training set. As we previously explained, these properties can be used to identify the GAN samples although they might be not perceptible with the naked eye.

## 4.3 BACH CHORALES

We investigate the properties of Bach chorales generated with the GAN framework and verify if they satisfy musical specifications learned from real data. Bach chorales are polyphonic pieces of music,

---

[4]Consider approximating a function with polynomials of increasing degrees.

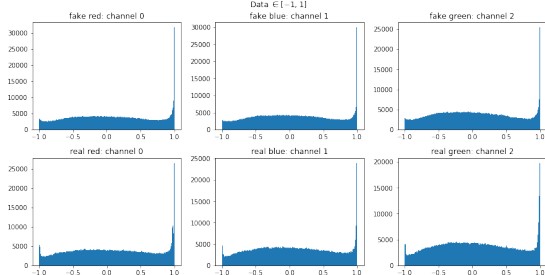
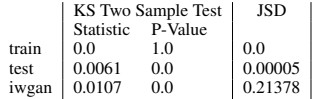

(a) Distribution of pixel intensities from real and fake samples      (b) Test statistics

Figure 6: CIFAR10 pixel distribution and test statistics

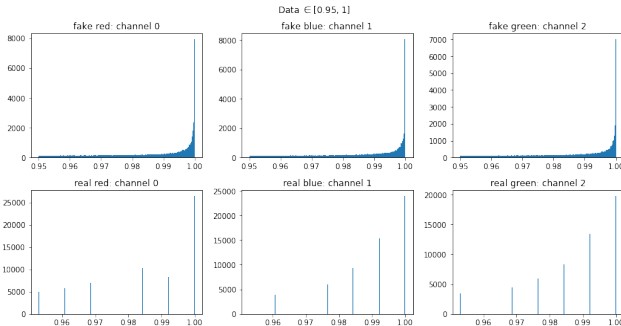

Figure 7: Distribution of pixel intensities from real and fake samples. The pixel distribution computed over fake samples is continuous and smoothly approximates the mode at 1.0

normally written for 4 or 5 voices, that follow a set of specifications or rules[5]. For example, a global specification could assert that only a set of durations are valid; a local specification could assert that only certain transitions between states (notes) are valid depending on the current harmony.

For this experiment, we convert the dataset of Bach chorales to piano rolls. The piano roll is a representation in which the rows represent note numbers, the columns represent time steps and the cell values represent note intensities. We compare the distribution of features computed over the training set, test set, GAN generated samples and a random baseline sampled from a Bernoulli distribution with probability equal to the normalized mean value of intensities in the training data. After scaling and thresholding, the intensities in the training and test data are strictly bi-modal and equal to 0 or 1. Figure 15 below shows training, test, IWGAN and Bernoulli samples, with modes on 0 and 1. Each dataset has approximately 1000 image patches.

Figure 8 shows a behavior that is similar to our previous MNIST experiments: the IWGAN asymptotically approximates the modes of the distribution of intensity values.

Following, we investigate if the generated samples violate the specifications of Bach chorales. We use these piano rolls to compute boolean Chroma (Peeters, 2004) features and to compute an empirical Chroma transition matrix, where the positive entries represent existing and valid transitions. The transition matrix built on the training data is taken as the reference specification, i.e. anything that is not included is a violation of the specification. Table 2 shows the number of violations given each dataset.

Although Figure 15 shows generated samples that look similar to the real data, the IWGAN samples have over 5000 violations, 10 times more than the test set! Violation of specifications is a strong evidence that fake samples do not come from the same distribution as the real data. Furthermore, to the trained ear the fake samples violate the style of Bach. We invite the readers to listen to them.

---

[5]The specifications define the characteristics of the musical style.

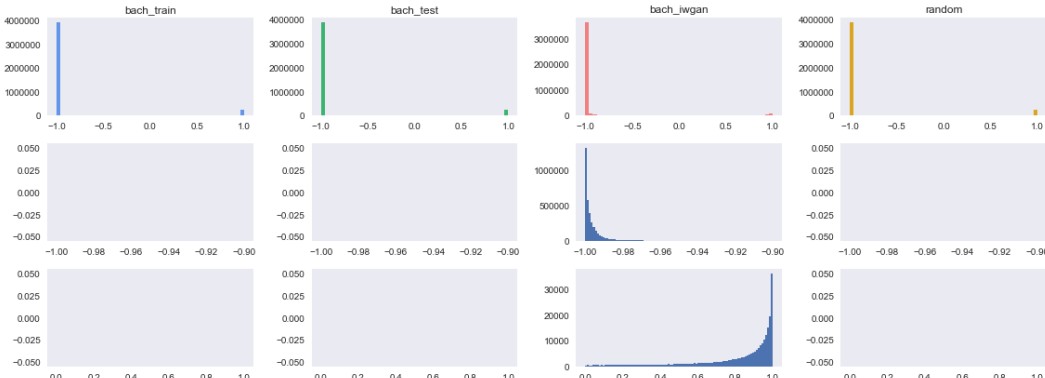

Figure 8: Bach Chorales intensity distribution

| | bach_train | bach_test | bach_iwgan | bach_bernoulli |
|---|---|---|---|---|
| Number of Violations | 0 | 429 | 5029 | 58284 |

Table 2: Number of specification violations with training data as reference.

In addition to experiments with Chroma features, we computed the distribution of note durations on the boolean piano roll described above. Figure 9a shows the distribution of note durations within each dataset. The train and test data are approximately bi-modal and, again, the improved WGAN smoothly approximates the dominating modes of the distribution. Table 9b provides a numerical comparison between datasets.

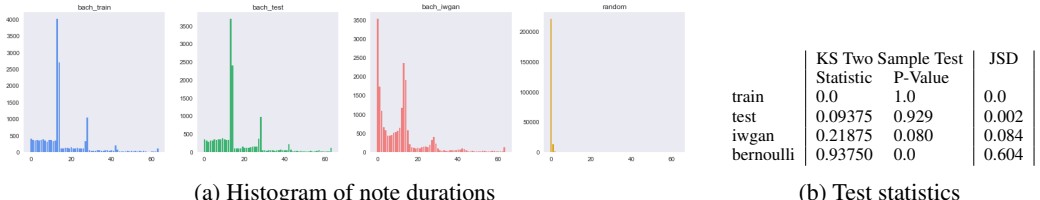

| | KS Two Sample Test | | JSD |
|---|---|---|---|
| | Statistic | P-Value | |
| train | 0.0 | 1.0 | 0.0 |
| test | 0.09375 | 0.929 | 0.002 |
| iwgan | 0.21875 | 0.080 | 0.084 |
| bernoulli | 0.93750 | 0.0 | 0.604 |

(a) Histogram of note durations                (b) Test statistics

Figure 9: Bach Chorales distribution of note durations and statistics

## 4.4 SPEECH

Within the speech domain, we investigate real and fake samples from Mel-Spectrograms. We divide the NIST 2004 dataset into training and test set, generate samples with the GAN framework and use a random baseline sampled from a Exponential distribution with parameters chosen using heuristics. The generated samples can be seen in Figure 16. We obtain the Mel-Spectrogram by projecting a spectrogram onto a mel scale, which we do with the python library librosa (McFee et al., 2015). More specifically, we project the spectrogram onto 64 mel bands, with window size equal to 1024 samples and hop size equal to 160 samples, i.e. frames are 100ms long. Dynamic range compression is computed as described in (Lukic et al., 2016), with $log(1 + C * M)$, where $C$ is the compression constant scalar set to 1000 and $M$ is the matrix representing the Mel-Spectrogram. Each dataset has approximately 1000 image patches and the GAN models are trained using DCGAN with the improved Wasserstein GAN algorithm.

Figure 10a shows the empirical CDFs of intensity values. Unlike our previous experiments where intensities (Bach Chorales) or pixel values (MNIST, CIFAR10) were linear and discrete, in this experiment intensities are continuous and compressed using the log function. This considerably reduces the distance between the empirical CDFs of the training data and GAN samples, specially around the saturating points of the tanh non-linearity, $-1$ and $1$ in this case.

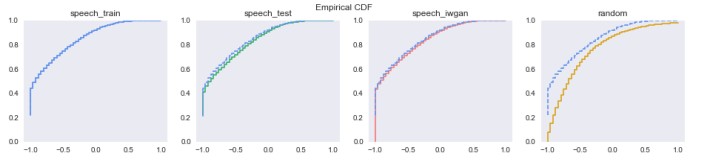

(a) Empirical CDF computed on intensities. Training data in blue and other datasets.

| | KS Two Sample Test | | JSD |
|---|---|---|---|
| | Statistic | P-Value | |
| train | 0.0 | 1.0 | 0.0 |
| test | 0.03685 | 0.0 | 0.00080 |
| iwgan | 0.22149 | 0.0 | 0.00056 |
| bernoulli | 0.36205 | 0.0 | 0.11423 |

(b) Test statistics

Figure 10: Empirical CDF and statistical tests of speech intensity

Table 10b shows a significant difference between the KS-Statistic of test samples and fake samples with respect to the training data. However, an adversary can manipulate the fake samples to considerably decrease this difference and still keep the high similarity in features harder to simulate such as moments of spectral centroid or slope.

Figure 11 shows the distribution of statistical moments computed on spectral centroids and slope. The distributions from different sources considerably overlap, indicating that the generator has efficiently approximated the real distribution of these features.

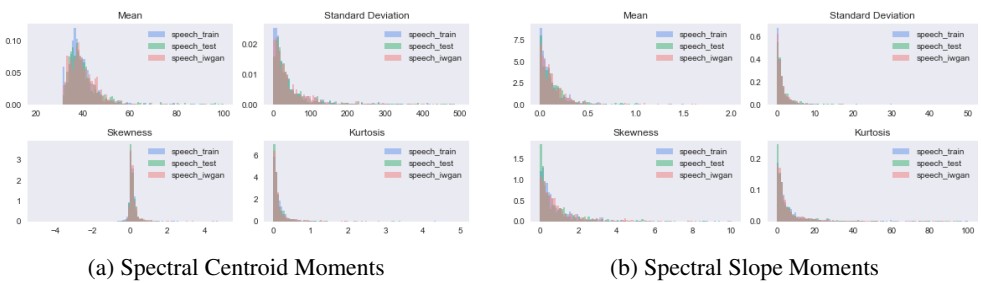

(a) Spectral Centroid Moments      (b) Spectral Slope Moments

Figure 11: Moments of spectral centroid (left) and slope(right)

## 5 CONCLUSIONS

In this paper we investigated numerical properties of samples produced with adversarial methods, specially Generative Adversarial Networks. We showed that fake samples have properties that are barely noticed with visual inspection of samples, namely the fact that fake samples smoothly approximate the dominating modes of the distribution due to stochastic gradient descent and the requirements of differentiability. We analysed statistical measures of divergence between real data and other data and the results showed that even in simple cases, e.g. distribution of pixel intensities, the divergence between training data and fake data is large with respect to test data. Finally, we mined specifications from real data and showed that, unlike test data, the fake data considerably violates the specifications of the real data.

In the context of adversarial attacks, these large differences in distribution and specially violations of specification can be used to identify data that is fake. In our results we show that, although some of the features used to learn specifications in this paper are weakly perceptually correlated with the content of the image, they certainly can be used to identify fake samples.

Although not common practice, one could possibly circumvent the difference in support between the real and fake data by training Generators that explicitly sample a distribution that replicates the support of the real data, i.e. 256 values in the case of discretized images. Conversely, one could mine specifications that are easy to learn from real data but hardly differentiable. These are topics that are not limited to GANs and remain to be explored in the larger domain of Verified Artificial Intelligence considered in (Seshia & Sadigh, 2016).

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

ACKNOWLEDGMENTS

We are thankful to Ryan Prenger and Kevin Shih for their feedback on this paper. We acknowledge NVIDIA for providing us with the Titan X GPU used in these experiments.

## APPENDIX A    SPECTRAL CENTROID AND SLOPE IMAGES

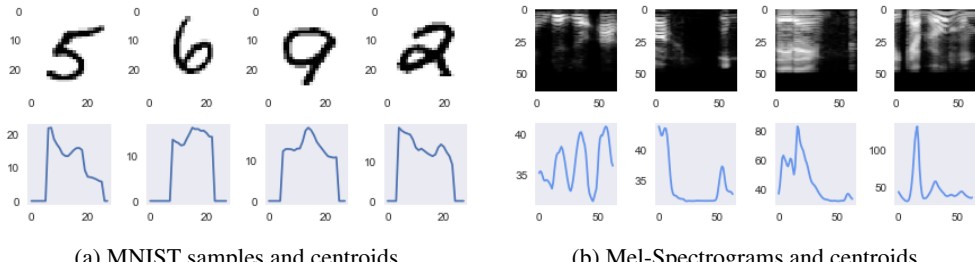

(a) MNIST samples and centroids

(b) Mel-Spectrograms and centroids

Figure 12: Spectral centroids on digits and Mel-Spectrograms

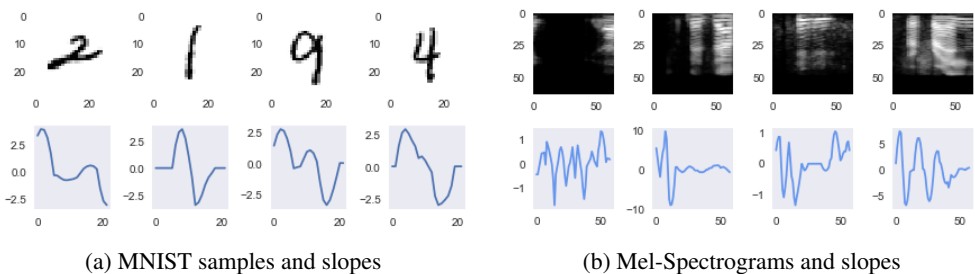

(a) MNIST samples and slopes

(b) Mel-Spectrograms and slopes

Figure 13: Spectral slopes on digits and Mel-Spectrograms

## APPENDIX B    MNIST IMAGES

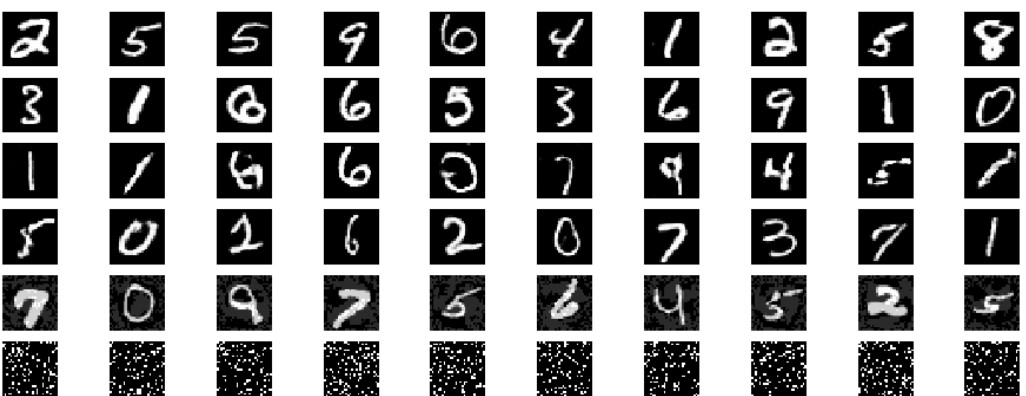

Figure 14: Samples drawn from MNIST train, test, LSGAN, IWGAN, FSGM and Bernoulli

## APPENDIX C    BACH CHORAL IMAGES

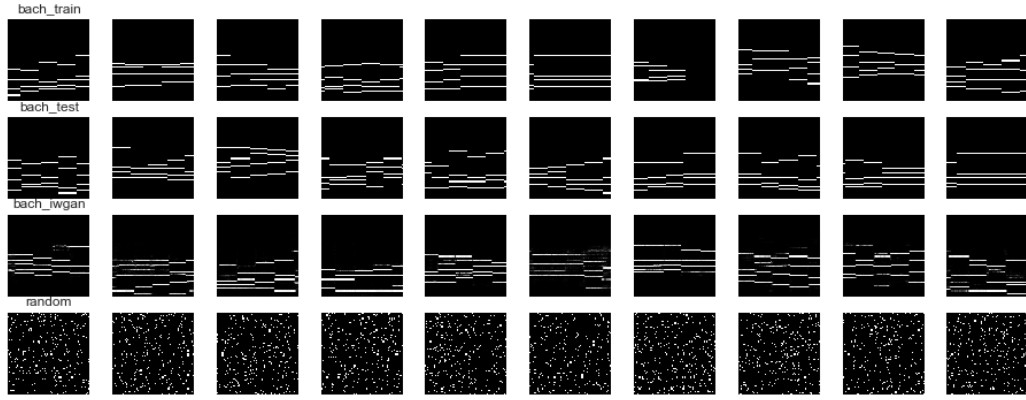

Figure 15: Samples drawn from Bach Chorales train, test, IWGAN, and Bernoulli respectively.

## APPENDIX D    MEL-SPECTROGRAM IMAGES

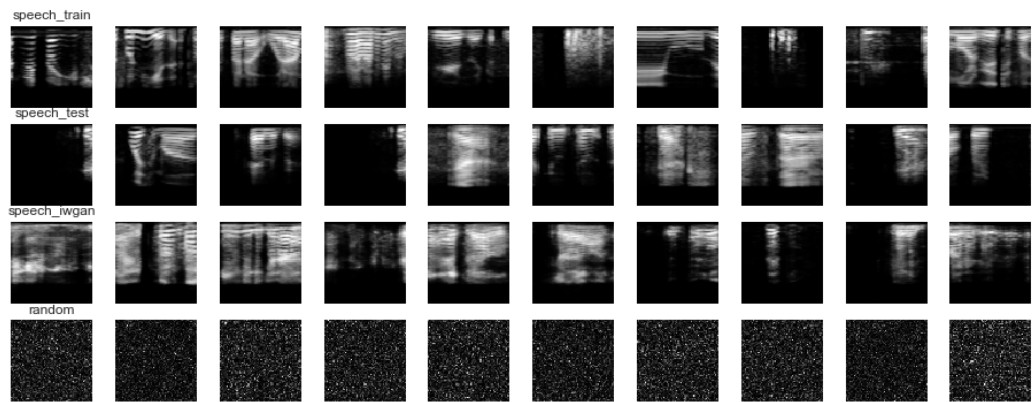

Figure 16: Samples drawn from NIST2004 train, test, IWGAN, and exponential respectively.

