# OpenReview forum: "TequilaGAN: How To Easily Identify GAN Samples"
_ICLR.cc/2019/Conference_

### Official Review · AnonReviewer2 · 2018-11-02
**Research that has some interesting observations, but needs to take the next step to have impact on the field**

**Rating:** 5
**Confidence:** 4

**Review:**

The primary purpose of this paper, from what I understand, is to show that fake samples created with common generative adversarial network (GAN) implementations are easily identified using various statistical techniques. This can potentially be useful in helping to identify artificial samples in the real world.

I think that the reviewers did an excellent job of probing into different statistical perspectives, such as looking at the continuity of the distribution of pixel intensities and the various higher moments of spectral data. I also must applaud the fact that they did not relegate themselves to image data, but branched out to speech and music data as well.

One of the first findings is that, with MNIST and CIFAR, the pixel intensities of fake samples are noticeably different when viewed from the perspective of a Kolgomorov-Smirnov Test or Jensen-Shannon Divergence comparison. This is an interesting observation, but less so than it would be if compared to something such as a variational autoencoder (VAE), which fits a KL distribution explicitly. IWGAN and LSGAN are using different metrics in their loss functions (such as Wassertein and least squares), and thus the result is not surprising or novel. I think if the authors had somehow shown how they worked their metrics into IWGAN or LSGAN to achieve better results, this could have been interesting.

Another observation the authors make is about the smoothness of the GAN distributions. This may not be so easily wrapped into the loss function, but it seems easily remedied as a post-processing step, or perhaps even a smoothing layer in the network itself. Nevertheless, this is an observation that I have not seen discussed in the literature so there is merit to at least noting the difference. It is confusing that on page 4, the authors state that they hypothesized that the smoothness was due to the pixel values themselves, and chose to alter the distribution of the original pixels in [0,1]. However, they state that in Figure 5, the smoothness remained "as expected." Did the authors misspeak here?

I found the music and speech experiments very interesting. The authors note that the synthetic Bach chorales, for instance, introduce many transitions that are not seen in the training and testing set of real Bach chorales. This, again, is interesting to note, but not surprising as the authors are judging the synthetic chorales on criteria for which they were not explicitly optimized. I do not believe these observations to be paper-worthy by themselves. However, the authors I believe have a good start on creating papers in which they specifically address these issues, showing how they can create better synthetic samples by incorporating their observations.

As to the writing style, there are many places where the writing is not quite clear. I would suggest getting an additional party to help proofread to avoid grammatical mistakes. I do not believe that the mistakes are so egregious as to impede understanding. However, it could distract from the importance on the authors future innovations if not corrected.

One last note. The title of the paper is "TequilaGAN: How to Easily Identify GAN Samples." This makes it seem as if the authors were introducing another type of GAN, like LSGAN or DCGAN. However, they are not. As a matter of fact, nowhere else in the paper is the word "TequilaGAN" mentioned. This title seems a bit sensational and misleading.

In the end, although I did find this paper to be an interesting read, I cannot recommend it for publication in ICLR.

----

Edit - November 29, 2018: Increasing my rating from a 4 to a 5 after discussion with the authors. Though their insights are not unknown, I think the authors are right in the fact that this is not explicitly discussed, at least not in the peer-review research with which I am familiar. But I don't think this by itself merits an ICLR publication.

---

> ### Author Response · Authors · 2018-11-19
> **1st reply to AnonReviewer2**
>
> Thank you for your valuable comments.
> We are hoping to exchange ideas and knowledge in this platform. Please let us know if any of your questions were not addressed here.
>
> "...if compared to something such as a variational autoencoder (VAE)..."
> We briefly speculated that fitting a KL distribution would circumvent such problems and immediately discarded this speculation because in the literature VAEs (and AEs) produce blurrier (smoother) samples than GANs do.
>
> Nonetheless, our paper explicitly focuses on GANs, hence we did not include comparisons with non-GANs models. Although comparing with other models is not within the scope of our paper, we believe and mentioned in the paper that the smoothness is due to the requirements of differentiation, etc...
>
> "in Figure 5, the smoothness remained..."
> To verify that the smoothness of fake samples was independent of the data itself, we binarized the data and showed that the smoothness remained, as expected.
>
> "...the authors are judging the synthetic chorales on criteria for which they were not explicitly optimized..."
> Having a good approximation to the entire distribution is a sufficient task to have a good approximation to features and their summary statistics, e.g transitions, computed on the entire distribution.
>
> In our paper, we show that the fake samples have considerably more specification violations than the test data. This is evidence that the generator has not properly learned the distribution of the data.
>
> "...The title of the paper is "TequilaGAN:..."
> The title is a wordplay between “To Kill A” and “Tequila”, hence TequilaGAN.
> We will add a footnote to clarify the wordplay.

---

### Official Review · AnonReviewer3 · 2018-11-02
**Very interesting topic!**

**Rating:** 6
**Confidence:** 5

**Review:**

The ability to detect generated samples is a very interesting topic and has recently triggered a lot of discussion. The paper is well written, easy to follow and the authors have done an extensive evaluation in a number of different GAN applications.


Comments:

1) Methodology - The GANs being evaluated were trained ignoring the statistics that the authors use to detect generated samples, and thus it is expected that there will be a difference. Have the authors attempted to include those in the loss function? It is a fair argument to say that some are not differentiable, but there are ways one could still incorporate them e.g. using REINFORCE. What do the authors thing about that?

2) Following (1), as far as the specifications are known one could train a GAN to fake them. What do the authors think about that? Do the authors think they could detect samples from that or such statistics could be used to devise better GAN losses?

3) The authors conclude that the smoothness and the differentiability of a loss function will always result to an inductive bias. However, that's an assumption given that there are no experiments trying to fake the detection, or experimenting with a large number of different architectures.

4) In CIFAR10 the authors state that the distributions of pixels was quite different specially in the values close to -1. Another way to see neural networks is as differentiable compressors. Many times, value distortions are correlated to the amount of compression. Have the authors seen differences in e.g. larger architectures?

5) On a last note, I would change the title as there is no proof that these tests / assumptions would hold for further research in the field. It would be great to show that the statistics used to detect GAN samples cannot be tricked.

Minor comments:
1) p.1 In the context of Verified Artificial IntelligenceSeshia [...] - needs a space.
2) p.3 Spectral centroid in 2 [...] - 2 -> Figure 2.
3) p.7 Figure 8 doesn't have a caption.
4) P.7 There are some figures above SPEECH without a figure number.
5) P.7 Reference to table 9b seems to be missing.

---

> ### Author Response · Authors · 2018-11-19
> **1st reply to AnonReviewer3**
>
> Thank you for your comments on our paper and suggestions including using REINFORCE or training GANs with extra loss terms related to specifications.
> We plan to use this platform to exchange ideas. Let us know if you have further suggestions or if any of your questions were not addressed.
>
> "1) Methodology ..."
> It is expected that without the summary statistics a priori specified, having a good approximation to the entire distribution is a sufficient task to have a good approximation to summary statistics (transformations) of the distribution. Hence, incorporating them should not be needed.
>
> The REINFORCE idea is neat and although SeqGAN authors have applied it for text generation, the space in which images and other data live is intrinsically continuous. Training on such data in a discrete manner is likely intractable.
>
> Also, Ian Goodfellow pointed out on reddit, GANs are only defined for real-valued data .
> https://www.reddit.com/r/MachineLearning/comments/40ldq6/generative_adversarial_networks_for_text/
>
> "2) Following (1), ..."
> We have very briefly explored the idea of adding to the GAN loss a term representing violation of specifications. However, we believe that simple reversible flow models with a single loss function, like OpenAI’s Glow, represent a better solution than hand-engineering loss terms. As we mentioned, having a good approximation of the entire distribution implies having a good approximation of specifications and summary statistics. Hopefully our research community will devote more time to such models.
>
> "3) The authors conclude..." and "4) In CIFAR10 the authors..."
> The inductive bias is a condition of parameter estimation in general and the inductive bias of models like neural networks is smooth interpolation. Consider, for example, approximating a complex non-linear function with a polynomials of increasing degree. The fact that we are still improving GAN models for image generation is evidence that the models are still smooth approximations of the real distribution.
>
> "5) On a last note, ..."
> Our research on this paper shows evidence that fake samples generated with GANs can be easily identified using summary statistics. We think it's fair to say that further research that shows evidence that the summary statistics “cannot be tricked” should be carried in further research.

---

### Official Review · AnonReviewer1 · 2018-11-06
**Continuous generators in GANs don’t capture discrete statistics?**

**Rating:** 4
**Confidence:** 4

**Review:**

The paper proposes statistics to identify fake data generated using GANs based on simple marginal (summary) statistics or formal specifications automatically generated from real data. The proposed statistics mostly boil down to the fact that continuous-valued generator neural networks can’t adequately generate data distributions that are topologically different from the distribution in the latent z-space. The differences in the summary data/ feature statistics and statistics corresponding to formal specifications between fake and real data are of the above nature.

This seems fairly obvious but I haven’t seen this property of GANs being exploited to distinguish between GAN generated and real-data

This property/ shortcoming of the generator is not surprising at all and has been acknowledged before. See, for example, the discussions in,

Khayatkhoei et al, Disconnected Manifold Learning for Generative Adversarial Networks, arXiv:1806.00880 (NIPS, 2018)

This has spurred various approaches to mitigate this shortcoming. See, for example,

Ben-Yosef and Weinshall, Gaussian Mixture Generative Adversarial Networks for Diverse Datasets, and the Unsupervised Clustering of Images, arXiv:1808.10356

Jang, Gu and Poole, Categorical Reparameterization with Gumbel-Softmax, arXiv:1611.01144 (ICLR 2017)

So, the fact that summary statistics predicated on discreteness of data or discreteness of their dependencies can distinguish GAN-generated data from real data is not surprising at all. In fact, in the paper itself, figure 10 and 11 show that for continuous data like speech, the proposed statistics are unable to distinguish between the fake and real data.

Beyond this, even though it's interesting, there isn’t enough contribution in the paper.  It would be great if the authors can extend this observation and show if such statistics can always be found and tricks like Gumbel-Softmax/ GMM-GANs etc are doomed to fail or if certain extentions of GAN architectures can handle such statistics.

Furthermore, the paper needs to provide more clarity/ clarifications about the following:

1.	Apart from formal specifications, the rest of the statistics are ad-hoc (e.g. the spectral centroid or the spectral slope which are just borrowed from the audio domain) – why should these be good for images?
2.	Training choices do not seem principled – GANs are trained till generated samples look like real samples. Why not use parameter settings and train to produced state of the art results with chosen architectures?
3.	Figure 1: Why does the CDF for the real data start in the middle of the figure? The figure purportedly shows bimodal 1D data for which the CDF should be a step function whereas the reference data has an inclined line. Why?
4.	Using the term ‘spectral’ (centroid and slope) for image features is misleading when spectral features are not computed.  Do these features capture spectral properties of images. How? Why are these good features?
5.	What does an “asymptotically converging activation function” mean?

6. Some typos need to be corrected. Figure # and caption (with dataset name) for Figure 6 needs to be provided, etc.

---

> ### Author Response · Authors · 2018-11-19
> **1st reply to AnonReviewer1**
>
> Thank you for sharing your comments and suggesting approaches that potentially would mitigate the shortcomings we described in our paper, including the GM-GAN and Categorical Reparameterization with Gumbel-Softmax.
> We would like to use this venue for an  exchange of ideas and knowledge and kindly ask you to let us know if further clarifications are needed.
>
> We first address your global comments and then address your numbered comments.
>
> "This has spurred various approaches..."
> Regarding the GM-Gan, the CIFAR10 generated images in the GM-GAN paper are blurry (smooth) and do not satisfy specifications, e.g. entities have deformed shapes. This is evidence that this approach fails to mitigate, even in small images, the mentioned shortcomings.
>
> Regarding the Categorical Reparameterization with Gumbel-Softmax approach, given that the space in which images and most data live is intrinsically continuous, training on such data in a discrete manner is very likely intractable.
>
> "So, the fact that summary statistics predicated..."
> Please look at Figure 10b and compare the KS Two-Sample Test statistics between train, test, iwgan and bernoulli. This statistic computed on Bernoulli samples is the farthest from train and the IWGAN statistic is much closer to the Bernoulli statistic than it is to train or test. Hence, the statistics can be used to distinguish between fake and real data.
>
> "1. Apart from formal specifications, the rest of the statistics..."
> Given that deep models for vision come from a computer vision feature engineering approach, it is expected that these deep models will succeed in learning the distribution of such features.
> Nonetheless, the spectral centroid can provide valuable information about the distribution of “noise” per column in an image. As a matter of fact, we  the distribution of mean mention that 'spectral centroids from fake data and Bernoulli data suggest that the fake images have noise that are also equally spatially distributed.'
>
> "2. Training choices do not seem principled..."
> As we describe in the paper, the models are trained following the standard practice: 'trained until the loss plateaus and the generated samples look similar to the real samples'. Note that in our related work section we comment on the disadvantages of using methods such as Gaussian Parzen Window fitting and Inception Scores to guide training.
>
> "3. Figure 1: Why does the CDF for the real data..."
> With the exception of the Bernoulli sampled data, named random in Figure 1, the data is mainly bimodal, not exclusively, with modes at 0 and 1. Hence the “inclined line”, different from the step function on the last image representing the samples drawn from a Bernoulli distribution.
>
> "4. Using the term ‘spectral’ (centroid and slope) for image features..."
> We will update the terms to avoid confusion. As we described, the spectral centroid is a good feature because it provides valuable information about the distribution of “noise” in the image. For example, in Figure 2 we should that the distribution of spectral centroid between Bernoulli and Fake data are similar. The spectral slope was used in the context of audio as a counter-example where it is harder to identify fake samples.
>
> "5. What does an “asymptotically converging activation..."
> We mean saturating activation functions. We will update the term in the paper.

---

> > ### Comment · AnonReviewer1 · 2018-11-26
> > **Clarifications on details, but larger questions remain!**
> >
> > Thanks for the clarifications. I'll use the original numbering as that in my review. Starting with the numbered clarifications sought:
> >
> > On numbered clarifications sought:
> >
> > (1) Statistics of GAN generated data - I feel 'feature engineering' is wrongly used here. The current practice is automatic representation (feature) learning from the training data. In the reply, the term "noise" seems to be used in a hand-waving fashion and doesn't really address the larger concern of the ad-hoc nature of the proposed statistics.
> > (2) Training choices - if standard practice is presented on extant models, then those exact details should be shared or better still, experiments should show that the results stand irrespective of the expected variability of practical choices made within the guidelines of the standard practice.
> > (3) Thanks for pointing out the oversight -- bimodal vs. mainly bimodal.  I still don't understand why the CDFs begin in the middle in the left two panels.
> > (4) The use of the term "spectral" for spatial features on images -- I don't see any updates in the revised draft. Was the draft revised?
> > (5) Considered resolved. please update the term.
> >
> > Larger Questions remain (let me number them):
> >
> > A. Ad hoc statistics: There is nothing principled in how the proposed statistics were derived or intuited nor in explanations of why GANs have difficulty in capturing these statistics. Why these statistics? Is there a larger family that can't be captured by GAN families? Why / why not? What makes it difficult for them to capture them?
> >
> > B. Discreteness vs continuity hypothesis: In my review, I had hypothesized that some of the observed differences arise from the fact that GANs are generating data in a continuous-valued space while the real-data is discrete. Similar is the case with formal specifications. The authors' response doesn't address this point at all - is this partly responsible for the differences observed in experiments? If not, why not?
> >
> > The fact that some of the referenced work produces blurry data is besides the point. I'm referring to them because they acknowledge the notion of discreteness that I talk about in the review and it'll be useful for the authors to understand that such problems are acknowledged (if not fully resolved) in the literature. This is ongoing research and there are other published approaches which could be used to improve data quality.
> >
> > As I pointed out in my review -- In the paper itself, figure 10 and 11 show that for continuous data like speech, the proposed statistics are unable to distinguish between the fake and real data.
> >
> > C. Utility of the idea of summary statistics/ formal specifications: Finally, an interesting question is if such specifications are available to GAN training, will the generated samples be high fidelity with respect to the specifications? Can additional statistics or formal specifications all be found which will be able to distinguish the generated samples from real?
> >
> > Note that the set of all moments of a PDF is equivalent to the PDF itself. Since, theoretically, a statistic can be seen as an estimator of functions of the moments of the PDF, GAN data should not be distinguishable from real (apart from all the glorious travails of practical ML :)), UNLESS the real data and the range of the GAN generator don't overlap (which is what is happening in the discreteness/ continuity case).
> >
> > So, the response addresses some minor points in my review, and the paper raises interesting questions, but doesn't go far enough to merit a publication.
> >
> > Hence, I stand by my original rating.

---

### Author Response · Authors · 2018-11-13
**1st reply to all reviewers**

We have uploaded the revised paper that addresses issues with writing style, layout, etc.

---

### Meta-Review · Area_Chair1 · 2018-12-15
**Interesting topic, not ready for publication.**

**Confidence:** 4
**Recommendation:** Reject

**Metareview:**

The paper points out a statistical properties of GAN samples which allows their identification as synthetic.

The paper was praised by one reviewer as well-written, easy to follow, and addressing an interesting topic. Another added that the authors did an excellent job of "probing into different statistical perspectives", and the fact that they did not confine their investigation to images.

Two reviewers leveraged the criticism that various properties discovered are not surprising given the loss functions and associated metrics as well as the inductive biases of continuous-valued generator networks. Tests employed were criticized as ad hoc, and reviewers felt that their generality was limited given their reduced sensitivity on certain modalities. (While Figure 10b is raised by the authors several times in the discussion, and the test statistics of samples are noted to be closer to the test data than to the random baseline, the test falsely rejects the null [p-value ~= 0.0] for non-synthetic test data.)

I would encourage the authors to continue this line of inquiry as it is overall agreed to be an interesting topic of relevance and increasing importance, however based on the criticisms of reviewers and the content of the ensuing discussion I do not recommend acceptance at this time.